# Analysis of Codon Usage Patterns of Six Sequenced *Brachypodium distachyon* Lines Reveals a Declining CG Skew of the CDSs from the 5′-ends to the 3′-ends

**DOI:** 10.3390/genes12101467

**Published:** 2021-09-23

**Authors:** Jianyong Wang, Yujing Lin, Mengli Xi

**Affiliations:** 1Key Laboratory of Forest Genetics and Biotechnology of Ministry of Education, Co-Innovation Center for Sustainable Forestry in Southern China, Nanjing Forestry University, Nanjing 210037, China; wangjy@njfu.edu.cn; 2Shanghai Center for Plant Stress Biology and Center for Excellence in Molecular Plant Sciences, University of Chinese Academy of Sciences, Shanghai 200032, China; yjlin@psc.ac.cn

**Keywords:** *Brachypodium distachyon*, codon usage bias, CG-skew

## Abstract

*Brachypodium distachyon*, a new monocotyledonous model plant, has received wide attention in biological research due to its small genome and numerous genetic resources. Codon usage bias is an important feature of genes and genomes, and it can be used in transgenic and evolutionary studies. In this study, the nucleotide compositions and patterns of codon usage bias were calculated using Codon W. Additionally, an ENC plot, Parity rule 2 and correspondence analyses were used to explore the major factors influencing codon usage bias patterns. The numbers of hydrogen bonds and skews were used to analyze the GC trend in the 5′-ends of the coding sequences. The results showed that minor differences in the codon usage bias patterns were revealed by the ENC plot, Parity rule 2 and correspondence analyses. The analyses of the CG-skew and the number of hydrogen bonds showed a declining trend in the number of cytosines at the 5′-ends of the CDSs (from the 5′-ends to the 3′-ends), indicating that GC may play a major role in codon usage bias. In addition, our results laid a foundation for the study of codon usage bias patterns in *Brachypodium* genus and suggested that the GC plays a major role in determining these patterns.

## 1. Introduction

*B. distachyon,* a new monocotyledonous model plant, has received wide attention in genetics and evolution research due to its small genome, short life cycle, easy transformation, self-proliferation and numerous genetic resources [1]. In recent decades, scientists have collected many inbred lines from Asia and Europe [2]. Since *B. distachyon* is self-fertile and not domesticated, researchers support the unaltered view that intra-species natural diversity among inbred lines depends on mutation and selection pressure, which can be reflected in the pattern of codon usage [3].

Owing to the redundancy of genetic codons, synonymous mutations have no effect on protein function or organismal fitness; therefore, they were initially considered “silent mutations” [4,5,6,7]. The sequencing of the organisms revealed that synonymous codons have a non-random distribution among genes or genomes, which is called codon usage bias; it is caused by a range of factors, including GC content, mutation, selection and codon position [8,9,10,11]. Codon usage bias influences transcription [12,13], protein translation [14,15] and gene expression levels [16,17]. Thus, the codon usage bias pattern provides important information for reconstruction vectors and for the improvement of expression levels, as well as for the revelation of plant–pathogen coevolutionary events and the improvement of host resistance [18,19,20]. Previous research has found that codon usage, frequently of high- and low-GC-content genes, is different in *B. distachyon* Bd21, and 27 codons were identified as optimal for evolution studies [8]. However, it is unclear whether the codon usage pattern is same in the other *B. distachyon* inbred lines. For example, Bd1-1, Bd18-1, Bd2-3, Bd29-1, Bd3-1 and Bd30-1 are all members of the “core inbred line set”, an important set of *B. distachyon* genetic stocks, which are often used to develop single seed descent-derived inbred lines. Hence, the objective of this study was to gather meaningful information on codon usage bias that could be used for the analysis of the evolution of *Brachypodium* in the post-genomic era.

In this study, we compared the codon usage bias patterns among six *B. distachyon* inbred lines, using overall GC content, the effective number of codons (ENC) and relative synonymous codon usage (RSCU). A neutrality plot analysis was performed and the optimal codons were determined. In addition, we investigated the relationships between the number of hydrogen bonds, the AT-skew, the CG-skew and the codon positions. The results provided insights into genetic modification and genome evolution that can be utilized to improve the genetic transformation efficiency and to understand the genetic variation of *Brachypodium*.

## 2. Materials and Methods

### 2.1. Coding Sequence (CDS) Data

The complete genomes of six *B. distachyon* lines, including Bd1-1, Bd18-1, Bd2-3, Bd29-1, Bd3-1 and Bd30-1 [3], were download from BrachyPan (https://phytozome-next.jgi.doe.gov/brachypan, accessed on 4 March 2021). In order to avoid sampling errors, all the CDSs were extracted in accordance with the following considerations: (1) the CDSs’ length should be the fold of three; (2) the CDSs should not be less than 300 bp; (3) the CDSs should contain the initiation codon (ATG) and termination codons (TAA, TAG or TGA); (4) the CDSs should not have one or more intermediate stop codons; and (5) the CDSs should be encoded by the longest primary transcripts [11]. All the filtered CDSs were used to analyze codon usage bias with Codon W 1.4.2 (https://sourceforge.net/projects/codonw/, accessed on 13 March 2021).

### 2.2. ENC

The ENC may be estimated by the degree of deviation among the synonymous codon usage patterns of the six genomes. Its value ranges between 20 (strong bias of codons) and 61 (all codons used randomly) [21,22]. The ENC value depends on GC3s, another index of codon usage that refers to the G-to-C ratio at the third position of one codon. The expected values were calculated using following formula:$$ENC=2+S+\frac{29}{S^{2}+{\left(1-S\right)}^{2}}$$
where *S* represents the GC3s value [22].

### 2.3. Parity Rule (PR2)

The PR2 plot is used to assess the influence of mutation pressure, natural selection or both on codon bias [23]. There is no deviation in the DNA under mutation and selection conditions when the content of A = T and G = C. The A3s/(A3s + T3s) and G3s/(G3s + C3s) were calculated and plotted to illustrate the relationship between mutation and selection pressures on each gene [24].

### 2.4. RSCU

The RSCU value, as proposed by Sharp and Li in 1986, is the ratio of the observed number of codons to the expected number of codons, and it was calculated using the following formula [25,26]:$$RSCU=\frac{g_{ij}}{\sum_{j}^{n_{i}}g_{ij}}n_{i}$$
where *g_ij_* represents the frequency of the *i*th codon for the *j*th amino acid and *n_i_* represents *n_i_* kinds of synonymous codons encoding the same amino acid. *RSCU* values of more than, less than and equal to 1.0 indicated positive, negative and no bias, respectively, in codon usage. Thus, codons with *RSCU* values > 1.0 were used more frequently than other codons.

### 2.5. Correspondence Analysis (COA)

The COAs of codon usage in the nuclear genomes of six *B. distachyon* lines were used to study the relationships between codon usage and factors based on RSCU in Codon W 1.4.2. In the COA analysis, the RSCU values of 59 codons, excluding ATG, TGG and three termination codons (TAA, TAG and TGA), were placed in a multi-dimensional space along 59 orthogonal axes to explain the data variation, and the correlation analysis between axis 1 and codon usage indices, including GC content, GC3s and ENC, were carried out in R (Version 4.0.2) [27].

### 2.6. Codon Position and Hydrogen Bonds

Villada’s method, which has been used in bacteria and archaea, was applied to analyze the relationship between codon position and hydrogen bonds in *B. distachyon* [28]. First, nucleotides in the CDSs of each genome were arranged in a matrix, with CDSs as rows and codons as columns. Next, the sequences were treated using Biostrings (https://bioconductor.org/packages/release/bioc/html/Biostrings.html, accessed on 3 April 2021) and SeqinR [29] to ensure that all the CDSs were left aligned from the 5′-end. Subsequently, the nucleotide composition was converted into the number of hydrogen bonds on the basis of adenine (thymine) having two hydrogen bonds and guanine (cytosine) having three hydrogen bonds. Finally, the mean number of hydrogen bonds at each codon position in the genomes of the six *B. distachyon* lines was calculated using Hmisc (https://rdocumentation.org/packages/Hmisc/versions/4.5-0, accessed on 3 April 2021), with a 95% confidence interval based on 1000 nonparametric bootstrap replicates.

### 2.7. CG-Skew and AT-Skew Values

The CG-skew and AT-skew values of the sequences from the six genomes were calculated in a sliding window of 100 bp with 1-bp steps, using the following formulae [30]:$$\mathrm{CG}-\mathrm{skew}=\frac{C-G}{C+G}\;\mathrm{and}\;\mathrm{AT}-\mathrm{skew}=\frac{A-T}{A+T},$$
where A, T, G and C represent the total numbers of theses bases in the 100-bp windows.

### 2.8. Optimal Codons

To determine the optimal codons, 5% of the total genes from each of the six *B. distachyon* lines located at the extreme right and left of axis 1 were regarded as high- and low-expression datasets, using the two-way chi-square contingency test, respectively. The codons were defined as optimal if they showed a significantly greater frequency of usage (*p* < 0.01) in the high-expression datasets than in the low-expression datasets [8].

## 3. Results

### 3.1. Nucleotide Base Composition in Genomes of Six B. distachyon Lines

To avoid sampling errors, six *B. distachyon* genome coding sequences (CDSs), Bd1-1, Bd18-1, Bd2-3, Bd29-1, Bd3-1 and Bd30-1, were filtered, resulting in 37261, 41810, 38148, 35932, 30621 and 36985 datasets, respectively. The nucleotide composition and codon usage bias indices of the filtered CDSs were estimated by Codon W 1.4.2, which included the frequencies of adenine, guanine, cytosine and thymine bases at the third positions of the codons (A3s, G3s, C3s and T3s, respectively), the GC content at the third positions (GC3s), the number of synonymous codons (L_sym), the total number of amino acids (L_aa), the general average hydrophobicity of the proteins (GRAVY) and the aromaticity of the proteins (Aromo), as shown using means and standard deviations of all the genes in Table 1. For the nucleotide composition, there were tiny differences at the third positions of the codons among the six inbred lines, conforming to C > G > T > A, which suggested that a similar codon composition was present in *B. distachyon*. The GC3s’ values were greater than 0.55, indicating that the six lines tended to use codons with G and C at the third codon positions. In Bd3-1, the G3s, C3s and GC3s were less than in the other five lines (Figure 1a).

GC content, as an indicator of the strength of the mutations in a biological evolutionary process, can reflect the composition of the DNA sequence. To investigate the nucleotide compositions of the six genomes, we calculated the GC content of each gene from datasets using Codon W 1.4.2. Next, the density curves of genes from each line were plotted, as shown in Figure 1b. The bimodal distribution of the GC content had mean values of 52.44–55.18%. The maximum GC content value appeared in the Bd1-1, at 55.18%, with a variance of 9.28, and the minimum value was 52.44%, in Bd3-1. Interestingly, the bimodal distribution of the six inbred lines illustrated that there were far more genes with low GC content than with high GC content.

The effective number of codons (ENC) was used to assess the heterogeneity of uniform codon usage, which may have reflected the codons’ degree of preference during the protein translation process. The ENC varied from 20 to 61. There was strong codon bias in genes with high expression levels when the ENC was less than 35, whereas an ENC greater than 50 indicated that the use of codons was random.

The ENC of the six lines were computed and a box plot was constructed to compare the differences in ENC, as shown in Figure 1c. The mean ENC ranged from 49.77 in Bd1-1 to 51.53 in Bd3-1. The ENC values from the all genomes were near 50, indicating that the codon usage bias was weak in *B. distachyon*.

### 3.2. ENC and GC3 Analyses

The GC3 was defined as the proportion of the GC content at the third positions of the codons to the total number of bases. ENC reflects the degree of preference for synonymous codon usage. An ENC-GC3s plot analysis is often used to analyze the relationship between nucleotide composition and codon bias. When the data point falls near the standard curve, the codon bias pattern is mainly determined by the mutation pressure rather than the external selection pressure. The similar ENC-GC3 plot distribution of the CDSs from the six lines is shown in Figure 2. Most points are in an ENC range of 25 to 61 and a GC3 range of 0.25 to 1.0. Although only a few CDSs were found close to the expected curve, a majority of the points deviated below the curve, with low ENC values (Figure 2), indicating that the codon usage biases of the six genomes were affected mainly by natural selection, mutation pressure and other factors (such as gene length and expression level). Furthermore, the codon biases were shown to have been shaped by the combined effects of mutation and selection pressures in previous studies of Poaceae [31,32], Orchidaceae [33] and Malvaceae [24].

### 3.3. PR2 Plot Analysis

Parity analysis is an efficient method for the assessment of the mutation pressure on the bases in the third positions of codons during the evolutionary process. In a PR2 plot, four regions are divided by two lines, which indicate A = T and G = C. The points represent the G3/(G3 + C3) and A3/(A3 + T3) values of CDSs distributed around the center position of a two-dimensional scatter plot. The PR2 plots of the six *B. distachyon* lines are shown in Figure 3. The values of the AT-biases were 0.46, 0.46, 0.45, 0.45, 0.45 and 0.45 in Bd1-1, Bd18-1, Bd2-3, Bd29-1, Bd3-1 and Bd30-1, respectively. However, the values of the GC-biases were 0.49, 0.49, 0.50, 0.50, 0.51 and 0.49, respectively. Consistently, the data of the different lines revealed similar patterns on the T/C-bias at the third position of the codons. As a whole, the unbalanced frequency between A/T and G/C suggested that the codon usage bias was not only affected by mutation but also by selection and other factors in *B. distachyon* inbred lines. Similar results in codon usage bias studies have been reported in Asteraceae [34] and Poaceae [35].

### 3.4. Correspondence Analysis (COA)

To investigate the main variation of the codon usage in the nuclear genomes of the six lines, correspondence analyses of all the genes in each genome were performed based on the RSCU values of 59 encoding codons (excluding Met, Trp and the three stop codons), as calculated by Codon W (Table 2). Axis 1, the primary explanatory axis, accounted for 45.14%, 44.49%, 43.42%, 42.08%, 44.91% and 39.94% of the overall variations of codon usage for Bd1-1, Bd18-1, Bd2-3, Bd29-1, Bd30-1 and Bd3-1, respectively.

The distribution of the GC content in monocotyledons genomes is divided into two regions, based on a GC content of 60% [8]. Thus, there were different kinds of genes associated with different GC contents; high-GC (≥60%), middle-GC (40–60%) and low-GC (≤40%) were plotted on the axes. Interestingly, gthe enes with low- and high-GC content were located to the left and right, respectively, in Bd1-1 and Bd2-3, but the opposite distribution pattern was found in Bd18-1, Bd29-1, Bd3-1 and Bd30-1 (Figure 4). A further analysis showed similar results between GC content and the gene position on the first axis, with an R2 value of 0.89–0.90 (Appendix A). We also investigated the distribution of synonymous codons on Axis 1 and 2. A similar phenomenon occurred in the four-type codons, with the G/C-ending codons being distinguished clearly from A/T-ending codons on Axis 1 and pyrimidines being distinguished from purines on Axis 2 (Appendix A). These results suggested that nucleotide composition was one of the major factors in the synonymous codon usage patterns of the six *B. distachyon* inbred lines.

### 3.5. RSCU Value Analysis

The RSCU value represents a quantitative relationship between the number of codons found and expected, reflecting the usage bias of different codons encoding the same amino acid. The RSCU values of each gene in the six lines were first calculated using Codon W. Next, 50 genes in each genome were selected randomly to form a dataset for the analysis of the RSCUs of the different lines (Figure 5). There were 27 codons with RSCU > 1 ending with C/G in the six *B. distachyon* inbred lines. As shown in Figure 5, the RSCU values of almost all the codons ending with G/C were greater than those ending with A/T. We also investigated the optimal codons in the six *B. distachyon* inbred lines. The 5% genes located at the two ends of the first axis in the COA analysis were selected from each genome to represent codon bias gene datasets (high-bias and low-bias datasets). In total, 27 optimal codons, including UUC, CUC, CUG, AUC, GUC, GUG, UAC, CAC, CAG, AAC, AAG, GAC, GAG, UCC, UCG, CCC, CCG, ACC, ACG, GCC, GCG, UGC, CGC, CGG, AGC, GGC and GGG, were found in all the lines. They mainly occurred in the high-bias gene datasets, according to chi-square tests. Thus, there was a high degree of unity among the codon usage preferences of the six lines, based on their RSCU values.

### 3.6. Hydrogen Bonds at Different Codon Positions

The positioning of synonymous codons in CDSs can dictate protein expression via many mechanisms, such as protein cotranslational folding and local translation efficiency [36]. Single-stranded DNA is useful for RNA polymerization during denaturation because of the strong effects on protein translation at the 5′-end regions of the CDSs [28]. Thus, to understand the influence of codon position on protein translation, we investigated the relationship between the number of hydrogen bonds and codon positions at the 5′-end regions of the CDSs. The average number of hydrogen bonds gradually declined along with the codon positions of the CDSs from the 5′-ends to the 3′-ends in all the genomes, reaching lower levels (7.35–7.45) after the 300th codon of the CDSs. For example, the average number of hydrogen bonds at the 334th codon in Bd3-1 was only 7.35 (Figure 6a).

Next, the codons were divided into two types, based on the number of hydrogen bonds: cheap codons (six or seven hydrogen bonds) and expensive codons (eight or nine hydrogen bonds). The expensive codons were mostly used at the 5′-ends of the CDSs in all the genomes, and the usage frequency of the expensive codons declined as the codons’ positions increased (Figure 6b). The key positions with equal usage frequencies of the expensive and cheap codons were different across the six lines. The small differences between the usage frequencies of the expensive and cheap codons were found after the 200–300th codons in Bd1-1, Bd18-1 and Bd30-1, where they were equal. After this position, the frequencies of the expensive codons were less than 50% and those of the cheap codons were greater than 50%. In Bd2-3, Bd29-1 and Bd3-1, the key positions appeared in the 100–200th codons and there was a greater difference between the usages of the expensive and cheap codons.

Thus, although the number of hydrogen bonds gradually declinedas the codon positions increased in the six *B. distachyon* lines, the choice of different synonymous codons in the different genomes may have affected hydrogen bonding.

### 3.7. AT- and CG-Skews

AT- and CG-skews are features of nucleotide usage and may be used to analyze the usage trends of the four bases in the genome, which are also useful to understanding mutation or selection on the transcription. We calculated the AT- and CG-skew values of each CDS in the six lines using a sliding window of 100 bps with 1-bp steps, using the following formulae: AT-skew = (A − T)/(A + T) and CG-skew = (C − G)/(C + G), where A, T, G and C represent the total bases in the 100-bp window.

As shown in Figure 7a, a significant downward trend in the CG-skew was observed in the six *B. distachyon* lines with different slopes. The greatest slope of the CG-skew appeared in Bd29-1 and the smallest appeared in Bd2-3. We also found the G and C contents were almost equal at the 5′-end regions of CDSs in Bd3-1, whereas slightly more Cs were used than Gs in the other five lines.

Although the AT-skew presented an overall upward tendency (up-down-up), unlike the CG-skew trend, the slopes differed among the lines (Figure 7b). The greatest increase occurred in Bd1-1, and the smallest increase appeared in Bd3-1, which stabilized at approximately 0.03 after 300 bp.

These results suggest that most cases followed a similar trend, with a preference for C at the 5′-ends and for G at the 3′-ends, suggesting that this is a conserved phenomenon among *B. distachyon* lines. However, the differences in the AT-skew trends appeared to be species-dependent, implying that they were affected by other factors.

## 4. Discussion

Owing to the redundancy of genetic codons, synonymous mutations in codons have no effect on protein function and organismal fitness, and therefore, were initially considered “silent mutations” [4,5,6,7]. Synonymous codons are non-randomly distributed among the genes or genomes of sequenced organisms, which is termed codon usage bias. The bias may be caused by various factors, including GC content, mutation, selection and codon position [8,9,10,11]. In this study, the CDSs from six *B. distachyon* inbred lines were used to analyze codon usage bias and the factors influencing the patterns. The bimodal distribution of the GC content and the negative correlation between the CG-skew and the number of hydrogen bonds in the codons positioned at the 5′-ends of the CDSs were detected in the six lines, indicating that GC content plays a major role in codon usage bias [37,38].

### 4.1. Multiple Factors Play a Role in GC Content Distribution in B. distachyon Inbred Lines

The GC content, as an indicator of the strength of the mutation pressure during the evolutionary process, may reflect the composition of a DNA sequence. The GC content of monocotyledonous plants has gradually increased and has been greater than that of dicotyledonous plants since their separation, approximately 200 million years ago [39]. In our study, the average GC content of the CDSs ranged from 52.53 to 57.68% in the six *B. distachyon* inbred lines. Interestingly, previous research showed a bimodal distribution of GC content in Bd21, which is similar to our results [8].

Currently, three hypotheses may be used to explain codon usage bias patterns: selection, mutation and GC-biased gene conversion [40]. The selection hypothesis states that thermal stability leads to a high GC content. However, base composition is also shaped by other factors. For instance, the specific functions of genes may determine the GC contents of gene regions, and then change the codon usage bias. The mutational theory states that GC mutations may be driven by high GC content, which, in turn, promotes the evolution of species. The GC-biased gene conversion hypothesis states that single-strand DNA could form heteroduplexes with homologous sequences during recombination. Mismatches in the heteroduplexes could be repaired by changing the nucleotide bias toward GC alleles. Muyle found that G or C is used preferentially and that GC-biased gene conversion is positively correlated with codon position in the rice genome [40]. These results indicated that the bimodal distribution of GC in Poaceae may be greatly affected by selection and GC-biased gene conversion [40]. However, we found that the bimodal pattern in Bd3-1 was different from the other five inbred lines, implying that other factors affected the pattern. A similar pattern would appear in all the lines if the recombination and GC-biased dynamics were mostly conserved; partial conservation may cause the differences in the intensity of the GC-biased gene conversion, as has been documented in polyploidy plants, such as Triticeae [41], Gossypium [42] and Nicotiana [43]. However, the *B. distachyon* inbred line Bd3-1 is a diploid plant; the pattern of GC-biased gene conversion may different with polyploidy plants. Just as Bd3-1 is different from other lines, *B. distachyon* contains many lines that may produce more interesting results. Our results could not represent the whole of *B. distachyon* or even the *Brachypodium*, since only six lines were investigated, a very small portion. Thus, the differences in GC-biased gene conversion need to be explored during the evolution of *Brachypodium* in the future.

### 4.2. Hydrogen Bonding at the 5′-Ends of CDSs Is Different between Plants and Bacteria

The positioning of synonymous codons in CDSs can dictate tprotein expression by many mechanisms, such as protein cotranslational folding and local translation efficiency [36]. Villada found that the number of hydrogen bonds in the 5′-ends of the CDSs gradually increased in the bacteria, achieving stability after the 15th codon, with the frequency of expensive codons being greater than 50% [28]. However, a contrary result was found in the *B. distachyon*. The number of hydrogen bonds gradually declined as the codon position increased, leading to equal frequencies between the expensive and cheap codons after the 200th to 300th codons. This trend was verified by the CG-skew. Villada implied that the coupling of transcription and translation requires more energy in the prokaryotes to avoid the premature termination of transcription when increasing the number of hydrogen bonds [28]. The *B. distachyon* species, with its high GC content, may feature different patterns, owing to the separation of translation and transcription in space and time. Interestingly, the frequency of the cheap codons was greater than that of the expensive codons after approximately the 110th codon in Bd3-1. Thus, the patterns of the two types of codons varied among all the lines, implying that other factors affected the distribution pattern. Choosing different synonymous codons could affect hydrogen bonding. Plants may use this advantage in order to smoothly reduce the energy requirement for unwinding the double-strand DNA molecule in CDSs. Just as Bd3-1 is different from other lines, there may be differences in energy requirements among lines. Hence, the problem needs to be further explored.

## 5. Conclusions

In this study, we systematically compared the differences in codon usage bias patterns among six *B. distachyon* lines. A declining trend in the number of cytosines in the CDSs from the 5′-ends to the 3′-ends was detected by CG-skew and the number of hydrogen bonds, indicating that GC may play a major role in codon usage bias. Further research is needed to understand the mechanism behind the GC content’s effect on codon usage, using multi-omics data from more inbred lines.

## Figures and Tables

**Figure 1 genes-12-01467-f001:**
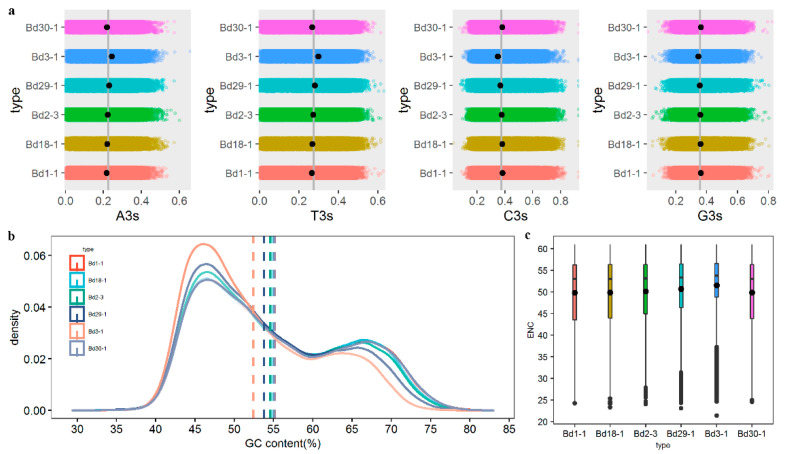
The nucleotide compositions and codon usages of six *B. distachyon* lines. (**a**) The distribution of four bases at the third positions of the codons; (**b**) The distribution of genes with different GC contents; (**c**) The box plot of ENC values from the six lines.

**Figure 2 genes-12-01467-f002:**
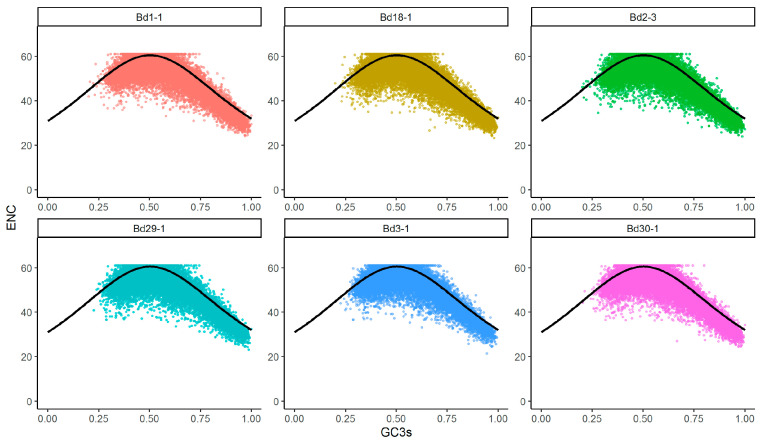
ENC-GC3 plots of the six *B. distachyon* genomes. The black solid lines indicate the expected ENC values.

**Figure 3 genes-12-01467-f003:**
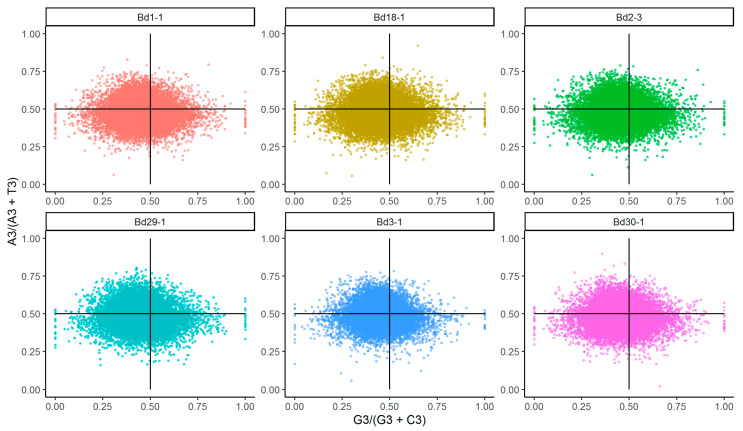
The Parity rule 2 (PR2) plots of the six *B. distachyon* lines.

**Figure 4 genes-12-01467-f004:**
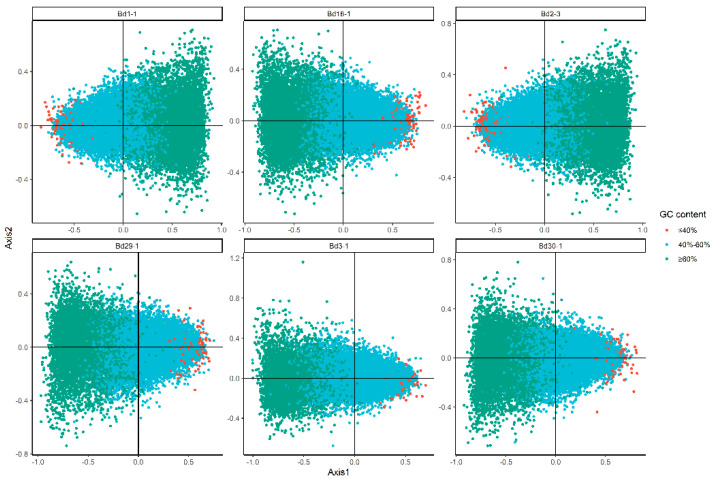
The correspondence analysis of the six *B. distachyon* inbred lines. The red, blue and green dots represent genes with GC content ≤ 40%, between 40–60% and ≥ 60%, respectively.

**Figure 5 genes-12-01467-f005:**
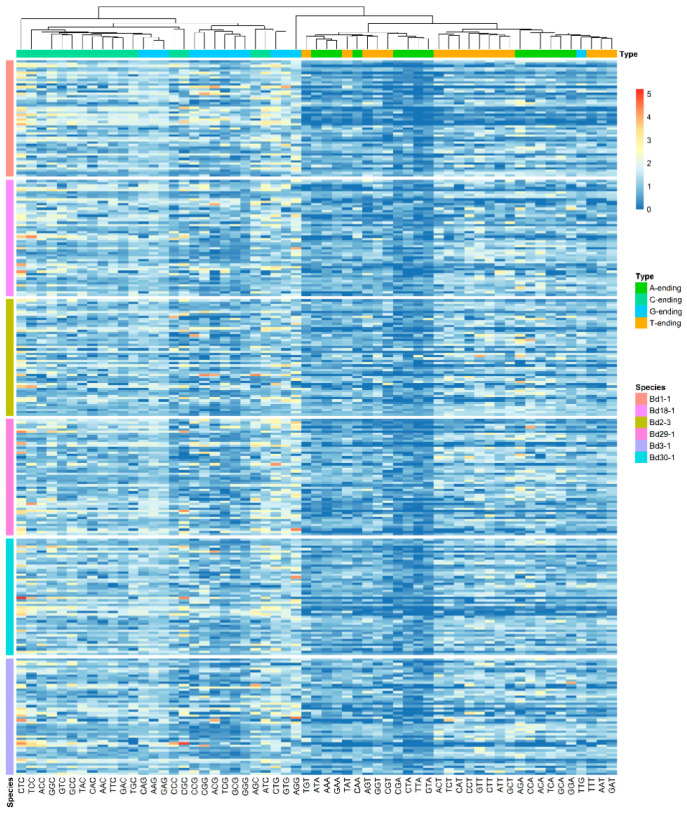
The relative synonymous codon usage (RSCU) values of the six genomes. A total of 50 genes were selected randomly from each of the six lines; these were plotted according to the RSCU values of 59 codons (Met, Trp and the three stop codons were omitted). The genes are represented in rows and the codons are represented in columns.

**Figure 6 genes-12-01467-f006:**
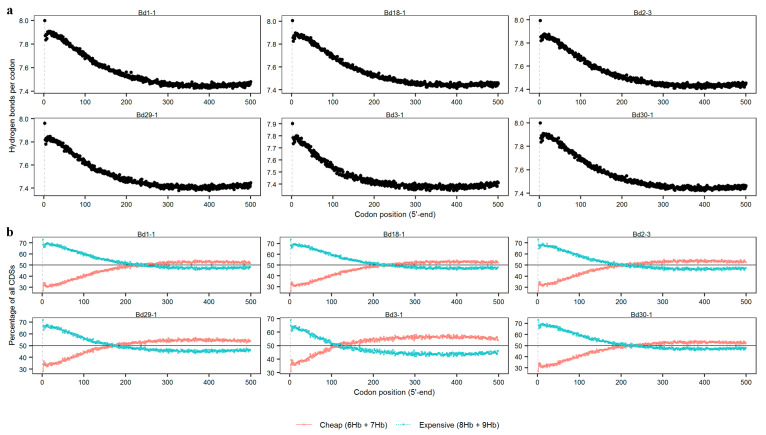
The relationship between the number of hydrogen bonds and codon position. (**a**) The hydrogen bonds per codon at the 5′-ends of the CDSs in the six lines; (**b**) The percentages of the two type codons at the 5′-ends of the CDSs in the six lines.

**Figure 7 genes-12-01467-f007:**
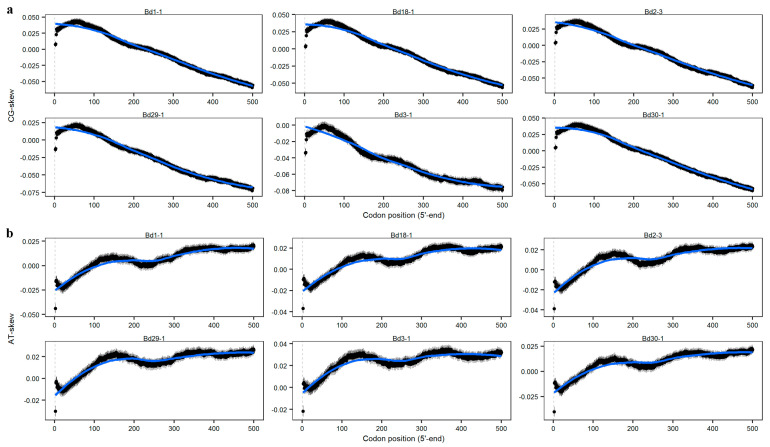
The relationships between both CG- and AT-skews and codon position. (**a**) CG-skews of nucleotides at the 5′-ends of the CDSs in the six lines; (**b**) AT-skews of nucleotides at the 5′-ends of the CDSs in the six *B. distachyon* lines.

**Table 1 genes-12-01467-t001:** The information pertaining to the six *B. distachyon* lines.

	Bd1-1	Bd18-1	Bd2-3	Bd29-1	Bd30-1	Bd3-1
Origin	Turkey	Turkey	Iraq	Ukraine	Spain	Iraq
Habit	winter	winter	spring	winter	winter	spring
Genome size	183 MB	269 MB	269 MB	253 MB	195 MB	170 MB
All GC *	45%	46%	46%	46%	45%	44%
Before filtered	42227	46621	45459	44278	42026	37234
After filtered	37261	41810	38148	35932	36985	30621
T3s	0.27 ± 0.13	0.27 ± 0.13	0.27 ± 0.13	0.28 ± 0.13	0.27 ± 0.13	0.30 ± 0.12
C3s	0.38 ± 0.14	0.38 ± 0.14	0.38 ± 0.14	0.37 ± 0.13	0.38 ± 0.14	0.35 ± 0.13
A3s	0.22 ± 0.11	0.22 ± 0.11	0.22 ± 0.11	0.23 ± 0.10	0.22 ± 0.11	0.24 ± 0.10
G3s	0.36 ± 0.10	0.36 ± 0.10	0.36 ± 0.09	0.35 ± 0.09	0.36 ± 0.10	0.35 ± 0.09
GC3s	0.60 ± 0.19	0.60 ± 0.18	0.59 ± 0.18	0.58 ± 0.18	0.60 ± 0.19	0.56 ± 0.17
L_sym	433.07 ± 303.94	439.45 ± 312.49	425.25 ± 302.44	421.47 ± 308.73	434.01 ± 305.26	424.25 ± 317.25
L_aa	449.64 ± 314.48	456.34 ± 323.33	441.66 ± 312.92	437.91 ± 319.52	450.68 ± 315.96	440.81 ± 328.13
Gravy	−0.27 ± 0.37	−0.27 ± 0.37	−0.27 ± 0.37	−0.27 ± 0.37	−0.27 ± 0.37	−0.28 ± 0.37
Aromo	0.08 ± 0.03	0.08 ± 0.03	0.08 ± 0.03	0.08 ± 0.03	0.08 ± 0.03	0.08 ± 0.03

* All GC represents the GC content of the genome.

**Table 2 genes-12-01467-t002:** The main variances of the four axes of the six lines.

Lines	Axis 1 (%)	Axis 2 (%)	Axis 3 (%)	Axis 4 (%)
Bd1-1	45.14	4.09	2.78	2.28
Bd18-1	44.49	4.18	2.84	2.29
Bd2-3	43.42	4.07	2.86	2.31
Bd29-1	42.08	3.99	2.95	2.40
Bd30-1	44.91	4.08	2.83	2.26
Bd3-1	39.94	3.93	3.03	2.46

## Data Availability

Not applicable.

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
