# Peer review of "Analysis of Codon Usage Patterns of Six Sequenced Brachypodium distachyon Lines Reveals a Declining CG Skew of the CDSs from the 5′-ends to the 3′-ends"

_genes, 2021, doi:10.3390/genes12101467_

Round 1
Reviewer 1 Report
Comments to the authors
Review for “Analysis of Codon Usage Patterns of Six Sequenced Brachypodium distachyon Lines Reveals a Decreasing Trend of GC content at the 5′-ends of the CDSs”
In this study, the authors compared the nucleotide composition of six different different Brachypodium distachyon Lines or accessions. They also analyzed the codon usage bias, GC-skew, hydrogen bonds along the CDS. There is not much study of GC-skew in eukaryotics, so the results are valuable. One of the lines seems to be quite different from other lines, which is very impressive. However, I do have problems with the writing of the manuscript and accuracy of presentation.
Major comments
- The title of the manuscript
I did not understand the title “a Decreasing Trend of GC content at the 5′-ends of the CDSs” before I read the manuscript and I still do not understand it even after I read the entire manuscript. This sentence has repeated many times in the manuscript but I am still sure what it means.
If we state “decreasing”, it implies there is a comparison. So please clarify what you are comparing with. Are you comparing 5′-ends with the remainder of the CDS?
GC content means the percent of G and C among all nucleotides. I am not super familiar with Brachypodium distachyon, but I am aware that in all grasses including Brachypodium distachyon, most genes demonstrate a negative GC gradient, which means GC content declines from 5′-end to 3’-end. If this is what you means, please state it correctly. However, I don’t think this trend has been directly revealed by this study. Moreover, I have to say this is well known and we do not need another study to confirm it. Based on the results I would guess it is the GC-skew index (GCSI, not the GC content) declines from 5′-end to 3’-end, that means C is overrepresented at the 5′-ends of the CDS.
- The background information about the six lines and Table 1
Row 129 – 131 “To minimize sampling errors, six Brachypodium distachyon genome coding sequences (CDSs), Bd1-1, Bd18-1, Bd2-3, Bd29-1, Bd3-1 and Bd30-1 were filtered, resulting in 37261, 41810, 38148, 35932, 30621 and 36985 datasets, respectively.”
As a mentioned, it is very impressive that there is an outliner (Bd3-1) among six lines. However if you compare the numbers in Table 1 with the statement above, nothing is consistent and there is no explanation.
“Bd1-1 Bd18-1 Bd2-3 Bd29-1 Bd30-1 Bd3-1
29501 39869 30012 36676 55997 44179”
Particularly, in Bd30-1, there are 55997 CDS, which is nearly twice of that of Bd1-1 (29501). This sounds very odd to me since they are all within the same species and they are all diploid. I am not really sure whether a Brachypodium distachyon should contain 56,000 genes which are many more than that in maize. Of course, if this is real, it is worth of more discussion.
So my suggestion is, make a table including all the background information about the six lines such as their geographic origin, their genome size or at least assembly size, the number of CDS before and after filtering, the GC content of the entire genomes. If you know how complete these assemblies are, that would also help.
Sometimes one gene have more than one model or CDS. In this case only one representative CDS should be taken.
In addition, please proof your text, tables, and Figures carefully and make sure everything is consistent. This is not difficult, and it is not good to make reviewers confused.
- The writing
The writing of the manuscript needs to be improved. Certainly language is an issue but organization and logic need more attention. The current results session is like a lab report that simply lists the results from all analyses there. My suggestion is to reorganize the results. For something that is just confirmative, minimize the description. For important or novel results, there should be more details including rationale and the biological implication. What is known, and what is still unclear. I consider the GC-skew part (Figure 7) quite interesting, especially, the Bd3-1 is so different from other lines. So you may consider to put more effort on this section and study why this line is different.
Having said that, please first double check whether such difference is an artifact of your analysis.
Since the number of CDSs from each line (that means the composition of genes is also different) is so different, it may help to just compare the syntenic genes from each line)
I understand that it is hard for non-native speaker to write. If language is an obstacle that prevents you from expressing fluently, you may consider writing using your native language first and then translate it into English. Just a suggestion. I personally consider the corresponding author should be responsible for making sure the manuscript is understandable before submission.
Minor comments:
- Row 66 “ The CDSs should contain the longest primary transcripts [11]”
It would better to state “The CDSs should be encoded by the longest primary transcripts [11]”
- Row 141 – 142 “The GC3 values were greater than 0.55, indicating that the six lines tend to use purine-rich codons more than pyrimidine-rich codons.”
GC3 has nothing to do with purine rich or pyrimidine rich because G is purine and C is pyrimidine, please rephrase it.
- Row 146 “Table 1. Nucleotide base composition of the genomes of six B.distachyon lines”
If I understand it correctly, this is the base composition of CDS, not the entire genome, please correct it.
- Row 253 – 254 “The specific arrangement of synonymous codons in a sequence affects protein folding…”
This sentence has appeared at least twice. I would like to point out that synonymous codons would not change amino acid sequences so it is impossible to affect protein folding. If you meant folding of transcripts, please change it.
- Row 315 – 316 “In this study, the CDSs from six species were used to analyze the codon usage bias and the factors influencing its patterns in B. distachyon lines.”
These are all B. distachyon so these are not six species, so please change to lines or accessions. Please also correct it in other places in the text.
Reviewer 2 Report
The authors aim to analyze codon usage patterns of six B. distachyon lines. The author carefully conducted the analysis. This paper is interesting. However, there are some comments.
- The rationale for selecting these six B. distachyon lines consisting of Bd1-1, Bd18-1, Bd2-3, Bd29-1, Bd3-1, and Bd30-1 was not given. Please give more explanation why the authors chose these six B. distachyon lines. There are other B. distachyon lines as well.
- There are other tools for analyzing codon usage. Why did the authors use Codon W 1.4.2? Why not ACUA, GCUA, EMBOSS, Jemboss, etc.
- Please add the limitation of your study.
- The paper is well-written with only few misspelling. Proofreading is suggested to refine the paper.
- At line 337, Please add the reference number after the author name.
Round 2
Reviewer 1 Report
Review for “Analysis of Codon Usage Patterns of Six Sequenced Brachypodium distachyon Lines Reveals a Decline Trend of the CG skew at the 5′-ends of the CDSs”.
This is a revision. The revised manuscript is indeed significantly improved. Particularly, I appreciate that authors clarified the impact of synonymous codons on protein folding and provides the reference. I was not aware of this study. Yet it would be good for authors to cite this reference in the text. In addition, the title and some of the statements within text are still problematic and they should be fixed.
I am copying some of the statements below:
“An interesting phenomenon was found that a decline trend of the number of the cytosines at the 5′-ends of the CDSs according to the results of CG-skew and the number of hydrogen bonds…”
“A decline trend of the number of the cytosine at the 5′-ends of the CDSs was detected by CG-skew and the number of hydrogen bonds,…”
Since the authors used “at the 5′-ends”, the title and the statements all imply that the number of Cs is lower at the 5′-ends than other regions, which is opposite to what their results are showing. As a result, this is incorrect and misleading. So my suggestion is to change the title to something like this:
“Analysis of Codon Usage Patterns of Six Sequenced Brachypodium distachyon Lines Reveals a Declining CG skew of the CDSs along the orientation of transcription”
Or simply
“Analysis of Codon Usage Patterns of Six Sequenced Brachypodium distachyon Lines Reveals a Declining CG skew of the CDSs from the 5′-ends to 3′-ends”.
Please change other places within the text accordingly.
Row: 31 – 32 “Since B. distachyon is self-fertility…”
It should be “Since B. distachyon is self-fertile…”
